# Factors influencing early sexual initiation among hill tribe youths in Chiang Rai Province, Northern Thailand: A community-based cross-sectional study

Ratipark Tamornpark[1,2], Tawatchai Apidechkul[1,2]*, Panupong Upala[1,2], Chalitar Chomchoei[1,2], Fartima Yeemard[1]

**1** Center of Excellence for The Hill Tribe Health Research, Mae Fah Luang University, Chiang Rai, Thailand, **2** School of Health Science, Mae Fah Luang University, Chiang Rai, Thailand

* tawatchai.api@mfu.ac.th

## Abstract

### Background

Early sexual initiation before the age of 15 years, prior to full physical and mental maturity, can lead to several problems, including sexually transmitted diseases. This study aimed to estimate the prevalence of and determine the factors associated with early sexual initiation among hill tribe youths in Thailand.

### Methods

A cross-sectional study was conducted to gather information using a validated questionnaire from hill tribe youths aged 15–24 years living in 30 selected hill tribe villages in Chiang Rai Province, Thailand. Data were obtained using self-reports in a private and confidential room. Chi-square tests and logistic regression were used to detect associations between variables at a significance level of α=0.05.

### Results

A total of 1,310 hill tribe youths participated in the study: 60.8% were females, 59.0% were aged 15–17 years, 34.0% were Akha, and 90.1% had no income. Substance use was reported as follows: 22.2% used alcohol, 14.3% smoked, and 2.6% used methamphetamine. A large proportion (79.8%) had poor knowledge of safe sex, and 33.5% had poor attitudes toward early sexual behaviors and safe sex. One-fourth (24.1%) had sexual experience. Of these, 42.0% were males, and 53.0% were female. Among those who had sexual experience, 9.8% had their first sexual initiation before 15 years. Of these, 81.1% were males, and 18.9% were females. The overall age at first sexual initiation was 16.7 years. The minimum age at first sexual initiation was 12 years for males and 13 years for females. Sex and age, including smoking, amphetamine, and heroin use behaviors, were associated with early sexual initiation in the univariate model. In the multivariable model, sex and age remained associated with early sexual initiation. Males were 7.30 times (95%

**Data availability statement:** The data underlying the results presented in the study are available from the supplement file attachment (S2).

**Funding:** The study was funded by The National Research Council of Thailand and Mae Fah Laung University, Thailand (Grant No.06/2560).

**Competing interests:** The authors have declared that no competing interests exist.

**Abbreviations:** CI, confidence interval; HIV, human immunodeficiency virus; IOC, item–objective congruence; UN, United Nations; WHO, World Health Organization

CI = 2.06–25.83) more likely to have experienced early sexual initiation than females were. Those aged 15–17 years were 8.31 times (95% CI = 1.68–41.07) more likely to have experienced early sexual initiation than those aged 21–24 years were.

## Conclusions

A large proportion of hill tribe youths have sexual experience, and one-tenth have experienced early sexual initiation before the age of 15 years. Specific knowledge on sexual education programs should be integrated into the high school study curriculum to protect early sexual initiation behaviors and safe sex, particularly for females. Family- and community-based interventions should also be strongly recommended for the implementation of sexual education training. The curriculum should focus on the impact of substance use and potentially early sexual initiation.

## Introduction

Youths constitute an age group between 15 and 24 years, as defined by the United Nations (UN) [1]. There is no exact definition for readiness to have sexual initiation. In addition to the use of a physical indicator of readiness for sexual initiation, many other factors need to be considered, depending on individuals' backgrounds, such as socioeconomic status [2], mental mutuality [3], and social acceptance [4]. Sexual initiation when one is not ready either physically or mentally will cause several problems in the future, such as sexually transmitted diseases [5,6] and school dropout among young parents [7]. Ideally, sexual intercourse occurs when individuals reach both physical and psychological maturity [8]. However, several factors, such as alcohol use [9], drug use [10], and social media use [11,12], influence early sexual initiation before individuals reach maturity, particularly in young people. The World Health Organization (WHO) defines early sexual initiation as having sexual intercourse before the age of 15 years; these behaviors vary across societies and frequently occur among youths [13]. Early sexual initiation could lead to several personal health and social problems in the future, representing a difficult issue to address. Youths subsequently become vulnerable to early sexual initiation due to various influencing factors in different groups of populations.

In 2019, the WHO reported that people with low education and those living in poverty were more likely to have early sexual initiation than people who had good education and high income [13]. The hill tribe people in Thailand have poor socioeconomic status [14,15]. In 2019, six main groups were identified: Akha, Lahu, Hmong, Yao, Karen, and Lisu. These groups had an overall population of approximately 4 million individuals in 16 provinces in Thailand [16]. Chiang Rai Province is located in the northernmost part of Thailand, bordering Myanmar in West, China in North, and Laos in East [17]. More than 350,000 hill tribe people live in 749 hill tribe villages in Chiang Rai Province [18]. The hill tribe people have their own lifestyle patterns [19], beliefs, perceptions [20], and acceptance of sexual intercourse [21].

The poor socioeconomic conditions of hill tribe people, including poor family and social relationships [19,22], could lead hill tribe youths to engage in early sexual initiation. Early sexual initiation in hill tribe youths, who have been clearly defined as vulnerable to sexually transmitted infections (STIs) [23], might lead to serious health problems from infections and unsafe abortion [24]. Given the family economic constraints of hill tribes [25], hill tribe youths also face limited access to care, which doubles their suffering. In addition, a large proportion of hill tribe youths have been directly and indirectly forced to work in cities [26], and new living circumstances could

lead them to experience early sexual initiation. In addition, some hill tribe individuals accept exposure to sexual activity early in life [18,20]. Currently, no scientific information is available regarding this issue. This study aimed to estimate the prevalence of and determine the factors associated with early sexual initiation or before the age of 15 years among hill tribe youths. The results could eventually be used to develop policies and public health interventions.

## Methods

### Study design and study population

A cross-sectional study design was used to estimate the prevalence and to determine the factors associated with early sexual initiation among hill tribe youths.

### Study setting

The study was conducted in hill tribe villages located in Chiang Rai Province, Thailand. In 2018, there were 749 hill tribe villages in Chiang Rai, comprising 316 Lahu villages, 243 Akha villages, 63 Yao villages, 56 Hmong villages, 36 Karen villages, and 35 Lisu villages. In 2018, a total of 42,177 hill tribe families lived in Chiang Rai Province [27].

### Study population

The study population was composed of hill tribe youths aged 15–24 years who lived in Chiang Rai Province. The target population consisted of six main groups: Akha, Lahu, Hmong, Yao, Karen, and Lisu.

The inclusion criteria were the hill tribe youths belonging to one of the six main hill tribes who lived in selected villages at the date of data collection. However, those who could not identify themselves as belonging to one of the six main tribes and were unable to communicate in Thai were excluded from the study.

### Study sample and sample size calculation

The study participants were 15–24 years old and were randomly selected from 30 hill tribe villages (5 villages for each tribe). The sample size was calculated on the basis of the standard formula for a cross-sectional study design [28]: n = [Z2a/2 P(1-P)]/e2, where n = sample size needed, Z = value from the standard normal distribution corresponding to the desired confidence level (Z = 1.96 for 95% CI), P = expected true proportion, and e = desired precision. Under the assumption of P = 0.15 [29] and e = 0.05, a total of 1,175 participants were required for the analysis. Considering a 10% error rate during the study, a total of 1,293 samples were required for the analysis.

### Research instruments

A validated questionnaire was used to collect data from the participants. The questionnaire was divided into four parts. In the first part, 13 questions were used to collect the general information of the participants, such as sex, age, and education. In the second part, 18 questions were used to collect information regarding health risk behaviors, such as drinking alcohol, smoking, and substance use. In the third part, 22 questions were used to collect information regarding the experience of early sexual initiation, namely, having their first sexual initiation before the age of 15 years, the age of their first sexual initiation, etc. In the last part, 20 questions were used to collect knowledge about safe sex and attitudes toward early sexual initiation behaviors and safe sex (S1 File).

Both the validity and the reliability of the questionnaire were determined before use. Three external experts (one public health expert, one epidemiologist, and one sexual and reproductive expert) in the field of sexual behavior were invited to validate the questions using via item-objective congruence (IOC) techniques. Those questions that scored less than 0.5 were excluded, questions that scored 0.50–0.69 were revised as comments, and questions that scored ≥ 0.70 were included in the questionnaire without revision.

Afterward, the questionnaire was piloted on 15 individuals with characteristics similar to those of the study sample in Huan Hin Fon and Sun Ti Suk Village, Mae Chan District, Chiang Rai Province. The purposes of performing a pilot test were to detect the feasibility of using the questions for the participants, to check the proper sequence of the questions throughout the questionnaire, and to detect the reliability of the last part of the questionnaire, which had a Cronbach's alpha of 0.71.

## Steps of data collection

A cluster random method was used to select 30 villages, with five villages representing each tribe, from 749 hill tribe villages located in the study setting: 316 Lahu villages, 243 Akha villages, 63 Yao villages, 56 Hmong villages, 36 Karen villages, and 35 Lisu villages. The district government officers granted permission to access villages before the village headmen were contacted. Afterward, the village headmen were contacted five days before meeting them in the village. Information about the research project, including its objectives and procedures, was provided to the village headman.

A list of the youths in the selected villages was asked for and obtained after approval was given by the village headman. All the youths in the village who met the study's criteria were invited to participate between November 2019 and March 2020. A further appointment was made to collect data from the target participants.

On the data collection date, all participants were provided information individually and asked for their cooperation by means of a signed informed consent form after a verbal explanation was provided. The participants completed the questionnaire independently in a private and confidential room in the village. For participants younger than 18 years, an informed consent form was also obtained from their parents. The whole process of completing the questionnaire lasted 25 minutes.

## Variables

The dependent variable was the experience of having sexual initiation before the age of 15 years, which was classified as "yes" and "no". Some key independent variables were categorized into proper forms. For example, education was classified into four categories: no education, primary school, high school, and vocational school. Variables related to substance use, such as smoking, alcohol use, and amphetamine use, were classified into two categories: "no" and "yes". The knowledge of safe sex section (10 items) was classified into three categories: poor (0–5 points), moderate (6–8 points), and good (9–10 points) [23]. There were three responses to each of the 10 attitudes questions, namely, agree, neutral, and disagree. The assessment included five positive questions, i.e., "An HIV-infected person can work with other people," and an additional five negative questions, i.e., "We should not use a bathroom with an HIV-infected person." For the positive questions, the following scores were given: agree (3 scores), neutral (2 scores), and disagree (1 score). However, for the negative questions, the following scores were given in the response: agree (1 score), neutral (2 scores), and disagree (3 scores). Those who scored 0–15 points were defined as having poor attitudes, those with scores ranging from 16–24 points had moderate attitudes, and those with scores ranging from

25–30 points were considered to have good attitudes toward early sexual initiation behaviors and safe sex.

## Statistical analysis

All completed questionnaires were coded, double-entered, and kept in secured files. The data were computed and analyzed using SPSS version 24 (SPSS, Chicago, IL). Descriptive statistics were used to explain the characteristics of the participants. Means and standard deviations (SDs) were used for the continuous variables, and proportions were used for the categorical variables.

Logistic regression was used to determine the associations between independent and early sexual initiation among hill tribe youths at a significance level of a = 0.05. At this step, we extracted only data from individuals who reported having had a sexual experience for incorporation into the model. During the analysis, a variable was selected into the model one by one until the model was fitted. The pseudo R2 values of the Cox–Snell R2, Nagelkerke R2, and Hosmer–Lemshow tests, which are chi-square tests, were determined to fit the statistical model. Age, tribe, and education were controlled as confounding factors in the model before the final findings were interpreted.

## Ethical approval and consent to participate

The research proposal, including the tools and procedures of data collection, was approved by the Mae Fah Luang University Research Ethics Committee on Human Research (REH-60030). All participants provided written informed consent. Participants younger than 18 years were required to ask their parents to sign an informed consent form before data collection began. The participants were compensated 200 baht ($6) each, as declared in the ethical approval process and application. All data collection processes were executed in a private and confidential room and performed in accordance with the relevant guidelines and regulations (Declaration of Helsinki).

## Results

In total, 1,310 participants were recruited for the study. The majority were females (60.8%), aged 15–17 years (59.0%), members of the Akha tribe (34.0%) and Buddhists (56.2%). A large proportion of the participants were single (94.3%), had graduated from high school (72.4%), and had no income (90.1%). Two hundred and eleven participants (16.1%) had experience working outside the village, and only 1.2% had experience working abroad (Table 1) (S2 File).

A total of 315 (24.1%) out of 1,310 reported that they had sexual experience, whereas 6.3% did not respond to the question. Among those who had sexual experience, 9.8% had their first sexual experience before the age of 15 years. Among those who experienced early sexual initiation, 87.1% were males (Table 1).

Comparisons of general characteristics among those with early sexual initiation, nonearly sexual initiation, those who have never had a sexual experience, and nonrespondents, several characteristics were found to exhibit statistically significant differences, except religion (Table 1).

One-fourth of the participants used alcohol (22.2%), 14.3% smoked, and 2.6% used methamphetamine. A large proportion (79.8%) had a poor level of knowledge of safe sex, and 33.5% had a poor level of attitudes toward early sexual initiation behaviors and safe sex (Table 2).

Several factors, including substance use behaviors, knowledge of safe sex, and attitudes toward early sexual initiation behaviors and safe sex, were found to be statistically significant

**Table 1. General characteristics of the participants.**

| Characteristics | Total | | Early sexual initiation | | | | Never had a sexual experience | | No answer | | χ² | p value |
|---|---|---|---|---|---|---|---|---|---|---|---|---|
| | | | Yes (≤14) | | No (≥15) | | | | | | | |
| | n | % | n | % | n | % | n | % | n | % | | |
| **Total** | **1,310** | **100.0** | **31** | **2.4** | **284** | **21.7** | **912s** | **69.6** | **83** | **6.3** | N/A | N/A |
| **Sex** | | | | | | | | | | | | |
| Male | 514 | 39.2 | 27 | 5.3 | 121 | 23.5 | 323 | 62.3 | 43 | 8.4 | 42.21 | <0.001* |
| Female | 796 | 60.8 | 4 | 0.5 | 163 | 20.5 | 629 | 74.0 | 40 | 5.0 | | |
| **Age** (years) | | | | | | | | | | | | |
| 15-17 | 773 | 59.0 | 19 | 2.5 | 69 | 8.9 | 650 | 84.1 | 35 | 4.5 | 286.15 | <0.001* |
| 18-20 | 392 | 29.9 | 8 | 2.0 | 120 | 30.6 | 228 | 58.2 | 36 | 9.2 | | |
| 21-24 | 145 | 11.1 | 4 | 2.8 | 95 | 65.5 | 34 | 23.4 | 12 | 8.3 | | |
| **Tribe** | | | | | | | | | | | | |
| Akha | 445 | 34.0 | 12 | 2.7 | 75 | 16.9 | 336 | 75.5 | 22 | 4.9 | 40.59 | <0.001* |
| Lahu | 204 | 15.6 | 4 | 2.0 | 65 | 31.9 | 120 | 58.8 | 15 | 7.4 | | |
| Hmong | 212 | 16.2 | 5 | 2.4 | 48 | 22.6 | 145 | 68.4 | 14 | 6.6 | | |
| Yao | 175 | 13.4 | 4 | 2.3 | 21 | 12.0 | 134 | 76.6 | 16 | 9.1 | | |
| Karen | 158 | 12.1 | 4 | 2.5 | 41 | 25.9 | 106 | 67.1 | 7 | 4.4 | | |
| Lisu | 116 | 8.9 | 2 | 1.7 | 34 | 29.3 | 71 | 61.2 | 9 | 7.8 | | |
| **Religion** | | | | | | | | | | | | |
| Buddhism | 736 | 56.2 | 16 | 2.2 | 169 | 23.0 | 496 | 67.4 | 55 | 7.5 | 9.25 | 0.131[a] |
| Christian | 566 | 43.2 | 15 | 2.7 | 115 | 20.3 | 409 | 72.3 | 27 | 4.8 | | |
| Islam | 8 | 0.6 | 0 | 0 | 0 | 0 | 7 | 87.5 | 1 | 12.5 | | |
| **Marital status** | | | | | | | | | | | | |
| Single | 1,235 | 94.3 | 28 | 2.3 | 223 | 18.1 | 910 | 79.7 | 74 | 6.0 | 178.53 | <0.001[a] |
| Married | 65 | 5.0 | 3 | 4.6 | 55 | 84.6 | 1 | 10.8 | 6 | 9.2 | | |
| Divorced | 10 | 0.7 | 0 | 0.0 | 6 | 60.0 | 1 | 40.0 | 3 | 30.0 | | |
| Characteristics | Total | | Early sexual initiation | | | | Never had a sexual experience | | No answer | | χ² | p value |
| | | | Yes (≤14) | | No (>15) | | | | | | | |
| | n | % | n | % | n | % | n | % | n | % | | |
| **Education** | | | | | | | | | | | | |
| No education | 45 | 3.4 | 2 | 4.4 | 17 | 37.8 | 18 | 40.0 | 8 | 17.8 | 139.13 | <0.001* |
| Primary school | 52 | 4.0 | 3 | 5.8 | 25 | 48.1 | 22 | 42.3 | 2 | 3.8 | | |
| High school | 949 | 72.4 | 24 | 2.5 | 130 | 13.7 | 744 | 78.4 | 51 | 5.4 | | |
| Vocational school | 246 | 18.8 | 1 | 0.4 | 105 | 42.7 | 122 | 49.6 | 18 | 7.3 | | |
| University | 18 | 1.4 | 1 | 5.6 | 7 | 38.9 | 6 | 33.3 | 4 | 22.2 | | |
| **Occupation** | | | | | | | | | | | | |
| Unemployed | 111 | 8.5 | 5 | 4.5 | 54 | 48.6 | 47 | 46.8 | 5 | 4.5 | 251.56 | <0.001* |
| Student | 1,048 | 80.0 | 19 | 1.8 | 140 | 13.4 | 826 | 84.8 | 63 | 6.0 | | |
| Farmer | 38 | 2.9 | 2 | 5.3 | 24 | 63.2 | 6 | 31.6 | 6 | 15.8 | | |
| Labor | 113 | 8.7 | 5 | 4.4 | 66 | 58.4 | 33 | 37.2 | 9 | 8.0 | | |
| **Having regular income** | | | | | | | | | | | | |
| No | 1,180 | 90.1 | 24 | 2.0 | 213 | 18.1 | 864 | 73.2 | 79 | 6.7 | 102.35 | <0.001* |
| Yes | 130 | 9.9 | 7 | 5.4 | 71 | 54.6 | 48 | 36.9 | 4 | 3.1 | | |
| **Work experience outside village** | | | | | | | | | | | | |
| No | 1,099 | 83.9 | 20 | 1.8 | 203 | 18.5 | 108 | 51.2 | 11 | 5.2 | 53.78 | <0.001* |
| Yes | 211 | 16.1 | 11 | 5.2 | 81 | 38.4 | 804 | 73.2 | 72 | 6.6 | | |

*(Continued)*

**Table 1.** (Continued)

| Characteristics | Total | | Early sexual initiation | | | | Never had a sexual experience | | No answer | | $\chi^2$ | p value |
|---|---|---|---|---|---|---|---|---|---|---|---|---|
| | | | Yes (≤14) | | No (≥15) | | | | | | | |
| | n | % | n | % | n | % | n | % | n | % | | |
| **Work experience abroad** | | | | | | | | | | | | |
| No | 1,294 | 98.8 | 29 | 2.2 | 273 | 21.1 | 910 | 70.3 | 82 | 6.3 | 26.88 | <0.001[a] |
| Yes | 16 | 1.2 | 2 | 12.5 | 11 | 68.8 | 2 | 12.5 | 1 | 6.3 | | |

*Significant at α =0.05, Fisher's exact test.

among those who had early sexual initiation, non-early sexual initiation, those who never had a sexual experience, and nonrespondents (Table 2).

Five hundred and four (38.5%) participants had boyfriends or girlfriends, and among them, 4.0% had more than one boyfriend or girlfriend. One-fourth (24.1%) of the participants had experienced sexual initiation (47.0% of males and 53.0% of females). Among those who had sexual initiation, 9.8% had sexual initiation before 15 years of age. The overall age at first sexual initiation was 16.7 years. After further examination, the average age at first sexual initiation among males was 16.0 years, with the minimum age at first sexual initiation being 12 years. In contrast, the average age at first sexual initiation among females was 17.4 years, with the minimum age being 13 years (Table 3).

Eighty-four participants (26.7%) had experienced sexual initiation with one-night stands (those with sexual experience without knowing their partner), 7.6% had sexual experience with a sex worker one year prior, and 19.1% regularly used alcohol or drugs prior to having sexual intercourse (Table 3).

In the comparisons of sexual behaviors between males and females, six factors were significantly different: number of boyfriends or girlfriends (p value=0.001), having sexual experience (p value<0.001), age at first sexual initiation (p value<0.001), having sexual experience with a one-night stand (p value<0.001), having sexual experience with a sex worker one year prior (p value=0.050), and having a regular partner (p value<0.001) (Table 3).

In the univariable model, five variables were associated with early sexual initiation: sex, age, smoking, methamphetamine use, and heroin use. Males were 9.09 times (95% CI=3.10–26.67) more likely to have experienced early sexual initiation than females were. Those aged 15–17 and 18–20 years were 6.54 times (95% CI = 2.13–20.08) and 4.13 times (95% CI = 1.71–9.93) more likely to have early sexual initiation than those aged 21–24 years, respectively. Those who smoked were 3.14 times (95% CI=1.47–6.67) more likely to have early sexual initiation than nonsmokers were. Those who used methamphetamine were 3.02 times (95% CI=1.03–8.85) more likely to have early sexual initiation than those who did not. Those who used heroin were 4.96 times (95% CI=1.17–20.93) more likely to have early sexual initiation than those who did not (Table 4).

In the multivariable model, sex and age were found to be associated with early sexual initiation. Males were 7.30 times (95% CI=2.06–25.83) more likely to have experienced early sexual initiation than females were. Those aged 15–17 years were 8.31 years (95% CI = 1.68–41.07) more likely to have early sexual initiation than those aged 21–24 years were (Table 4).

## Discussion

The hill tribe youths in Thailand used different substances, such as alcohol, tobacco, and amphetamines, and some of them used substances before sexual initiation. One-fourth (24.1%) of the participants had a sexual experience, whereas 6.3% did not indicate whether they had sexual experience or not. Among those who had sexual experience, 9.8% had their first sexual experience before the age of 15 years, and 87.1% were males. Males and younger

**Table 2. Substance use behaviors, knowledge of safe sex, and attitudes toward early sexual initiation behaviors and safe sex.**

| Characteristics | Total | | Early sexual initiation | | | | Never had a sexual experience | | No answer | | χ² | p value |
|---|---|---|---|---|---|---|---|---|---|---|---|---|
| | | | Yes (≤14) | | No (>15) | | | | | | | |
| | n | % | n | % | n | % | n | % | n | % | | |
| **Smoking** | | | | | | | | | | | | |
| No | 1,123 | 85.7 | 15 | 1.3 | 212 | 18.9 | 825 | 73.5 | 71 | 6.3 | 80.50 | <0.001* |
| Yes | 187 | 14.3 | 16 | 8.6 | 72 | 38.5 | 87 | 46.5 | 12 | 6.4 | | |
| **Alcohol use** | | | | | | | | | | | | |
| No | 1,019 | 77.8 | 13 | 1.3 | 162 | 15.9 | 777 | 76.3 | 67 | 6.6 | 123.18 | <0.001* |
| Yes | 291 | 22.2 | 18 | 6.2 | 122 | 41.9 | 135 | 46.4 | 16 | 5.5 | | |
| **Methamphetamine use** | | | | | | | | | | | | |
| No | 1,276 | 97.4 | 26 | 2.0 | 267 | 20.9 | 901 | 70.6 | 82 | 6.4 | 30.63 | <0.001[a] |
| Yes | 34 | 2.6 | 5 | 14.7 | 17 | 50.0 | 11 | 32.4 | 1 | 2.9 | | |
| **Heroin use** | | | | | | | | | | | | |
| No | 1,287 | 98.2 | 28 | 2.2 | 278 | 21.6 | 899 | 69.9 | 82 | 6.4 | 7.99 | <0.036[a] |
| Yes | 23 | 1.8 | 3 | 13.0 | 6 | 26.1 | 13 | 56.5 | 1 | 4.3 | | |
| **Crystal methamphetamine use** | | | | | | | | | | | | |
| No | 1,290 | 98.5 | 29 | 2.2 | 278 | 21.6 | 901 | 69.8 | 82 | 6.4 | 5.62 | <0.088[a] |
| Yes | 20 | 1.5 | 2 | 10.0 | 6 | 30.0 | 11 | 55.0 | 1 | 5.0 | | |
| **Level of knowledge of safe sex** | | | | | | | | | | | | |
| Poor | 1,045 | 79.8 | 25 | 2.4 | 191 | 18.3 | 758 | 72.5 | 71 | 6.8 | 37.01 | <0.001[a] |
| Moderate | 192 | 14.6 | 5 | 2.6 | 74 | 38.5 | 105 | 54.7 | 8 | 4.2 | | |
| Good | 73 | 5.6 | 1 | 1.4 | 19 | 26.0 | 49 | 67.1 | 4 | 5.5 | | |
| **Level of attitudes toward early sexual initiation behaviors and safe sex** | | | | | | | | | | | | |
| Poor | 439 | 33.5 | 13 | 3.0 | 95 | 21.6 | 331 | 75.4 | 41 | 9.3 | 13.47 | <0.036* |
| Moderate | 574 | 43.8 | 13 | 2.3 | 118 | 20.6 | 443 | 77.2 | 26 | 4.5 | | |
| Good | 297 | 22.7 | 5 | 1.7 | 71 | 23.9 | 221 | 74.4 | 16 | 5.4 | | |

*Significant at α =0.05, Fisher's exact test.

ages were identified as the key contributors to early sexual initiation among hill tribe youths in Thailand.

In our study, males (12 years old) were found to have an earlier age at first sexual initiation than females (13 years old). However, the overall age at first sexual initiation among hill tribe youths was 16.7 years. Specifically, the age was 16.0 years among males and 17.4 years among females. A study in South Africa among school children between grades 9 and 11 [30] reported that 50.4% of males and 30.1% of females had sexual experience. The median ages at first sexual initiation for males and females were 16.5 years and 17.5 years, respectively. A study in Ethiopia [31] reported that the mean age at sexual initiation among youths who had experienced sexual initiation was 15.4 years, whereas the corresponding age was 16.2 years among hill tribe youths in Thailand. A study in Ireland [32] reported that a greater proportion of girls (22.8%) than boys (13.4%) had sexual initiation prior to 14 years of age, in contrast to the findings for hill tribe youths. A study conducted among children who were attending school in the central region of Thailand [33] reported no sex-based differences in the proportion of early sexual experience among the youths (24.3% males and 24.0% females); however, 35.0% of the youths had more than one sexual partner. This rate is lower than that of hill tribe youths, who are sexually active (30.4%), with a greater proportion of males than females having sexual initiation. In another study [34] conducted among high school children aged 16–20 years living in rural northern areas of Thailand, 29.5% had experienced sexual initiation,

**Table 3. Comparisons of sexual behavior between sexes.**

| Characteristics | Total n (%) | Male n (%) | Female n (%) | χ² | p value |
|---|---|---|---|---|---|
| Total | 1,310 | 514 | 796 | N/A | N/A |
| **Having boyfriend or girlfriend** | | | | | |
| No | 806 (61.5) | 315 (39.1) | 491 (60.9) | 0.21 | 0.885 |
| Yes | 504 (38.5) | 199 (39.5) | 305 (60.5) | | |
| **Number of boyfriends or girlfriends** | | | | | |
| One | 484 (96.0) | 184 (38.0) | 300 (62.0) | 10.99 | 0.001* |
| More than one | 20 (4.0) | 15 (75.0) | 5 (25.0) | | |
| **Having sexual experience** | | | | | |
| No | 912 (69.6) | 323 (35.4) | 589 (64.6) | 19.01 | <0.001* |
| Yes | 315 (24.1) | 148 (47.0) | 167 (53.0) | | |
| No answer | 83 (6.3) | 43 (51.8) | 40 (48.2) | | |
| **Age at first sexual initiation (years)** | | | | | |
| ≤14 | 31 (9.8) | 27 (87.1) | 4 (12.9) | 24.59 | <0.001* |
| ≥ 15 | 284 (90.2) | 121 (42.6) | 163 (57.4) | | |
| *Mean overall = 16.7 (SD = 2.0); Median = 16, Mean in males = 16.0 (SD = 1.92); Mean in females = 17.4 (SD = 1.90)* | | | | | |
| **Having sexual experience with a one-night stand** | | | | | |
| No | 231(73.3) | 94 (40.7) | 137 (59.3) | 13.76 | <0.001* |
| Yes | 84 (26.7) | 54 (64.3) | 30 (35.7) | | |
| **Having sexual experience with sex worker one year prior** | | | | | |
| No | 291(92.4) | 131 (45.0) | 160 (55.0) | 5.93 | 0.015* |
| Yes | 24 (7.6) | 17 (70.8) | 7 (29.2) | | |
| **Having regular partner** | | | | | |
| No | 191 (63.6) | 107 (56.0) | 84 (44.0) | 15.9 | <0.001* |
| Yes | 124 (36.4) | 41 (33.1) | 83 (66.9) | | |
| **Alcohol or drug use prior having sexual initiation** | | | | | |
| No | 284 (76.4) | 130 (45.8) | 154 (54.2) | 1.80 | 0.406 |
| Sometime | 18 (4.5) | 10 (55.6) | 8 (44.4) | | |
| Yes | 13 (19.1) | 8 (61.5) | 5 (38.5) | | |

*Significant at α =0.05.

including 35% among males and 25.5% among females. Tangmunkongvorakul et al. [34] reported that the average age at first sexual initiation was 15.8 years, specifically, 15.9 years for males and 15.8 years for females. Compared with Thai youths, hill tribe youths are younger at first sexual initiation, which might be due to the culture of hill tribes that encourages getting married early to have a child for help in their family work. This is clearly supported by a study by Detpitukyon et al., who reported that Lahu people need to marry early to support their family [35].

Early sexual behavior in hill tribes was more common among males than females. This finding coincides with a longitudinal analysis conducted among Latin American youths, which reported that males had greater odds of having early sexual initiation than females did [36]. Gazendam et al. [37] reported that boys have a greater chance of having early sexual initiation than girls in Canada. A study in Nigeria [38] reported that being male was a factor associated with early sexual initiation among high school students. A study among vocational students in northern Thailand reported that male sex was one of the factors associated with early sexual initiation, particularly among those who used alcohol [39]. The gender

**Table 4. Univariable and multivariable analyses of factors associated with early sexual initiation among participants.**

| Factors | Early sexual initiation | | OR | 95% CI | p value | AOR | 95% CI | p value |
|---|---|---|---|---|---|---|---|---|
| | Yes<br>n (%) | No<br>n (%) | | | | | | |
| **Total** | **31(9.8)** | **284 (90.2)** | **N/A** | **N/A** | **N/A** | **N/A** | **N/A** | **N/A** |
| **Sex** | | | | | | | | |
| Male | 27 (18.2) | 121 (81.8) | 9.09 | 3.10–26.67 | <0.001* | 7.30 | 2.06–25.83 | 0.002* |
| Female | 4 (2.4) | 163 (97.6) | 1.00 | | | 1.00 | | |
| **Age** (years) | | | | | | | | |
| 15–17 | 19 (21.6) | 69 (78.4) | 6.54 | 2.13–20.08 | 0.001* | 8.31 | 1.68–41.07 | 0.009* |
| 18–20 | 8 (6.2) | 120 (93.8) | 4.13 | 1.71–9.93 | 0.002* | 2.33 | 0.47–11.41 | 0.270 |
| 21–24 | 4 (4.0) | 95 (96.0) | 1.00 | | | 1.00 | | |
| **Tribe** | | | | | | | | |
| Akha | 12 (13.8) | 75 (86.2) | 1.00 | | | 1.00 | | |
| Lahu | 4 (5.8) | 65 (94.2) | 0.38 | 0.11–1.25 | 0.112 | 0.57 | 0.11–2.78 | 0.493 |
| Hmong | 5 (9.4) | 48 (90.6) | 0.65 | 0.21–1.96 | 0.446 | 1.26 | 0.26–6.15 | 0.769 |
| Yao | 4 (16.0) | 21 (84.0) | 1.19 | 0.34–4.07 | 0.781 | 1.31 | 0.23–7.34 | 0.758 |
| Karen | 4 (8.9) | 41 (91.1) | 0.61 | 0.18–2.01 | 0.417 | 0.96 | 0.22–4.20 | 0.965 |
| Lisu | 2 (5.6) | 34 (94.4) | 0.36 | 0.07–1.73 | 0.206 | 0.40 | 0.05–3.08 | 0.380 |
| **Religion** | | | | | | | | |
| Buddhism | 16 (8.6) | 169 (91.4) | 1.00 | | | 1.00 | | |
| Christianity | 15 (11.5) | 115 (88.5) | 1.37 | 0.65–2.89 | 0.398 | 1.01 | 0.30–3.24 | 0.999 |
| **Education** | | | | | | | | |
| No education | 2 (10.5) | 17 (89.5) | 1.00 | | | 1.00 | | |
| Primary school | 3 (10.7) | 25 (89.3) | 1.02 | 0.15–6.76 | 0.984 | 1.77 | 0.12–25.24 | 0.673 |
| ≥High school | 26 (9.7) | 242 (90.3) | 0.91 | 0.20–4.17 | 0.907 | 0.81 | 0.07–9.38 | 0.869 |
| **Occupation** | | | | | | | | |
| Student | 19 (11.9) | 140 (88.1) | 1.00 | | | 1.00 | | |
| Unemployed | 5 (8.5) | 54 (91.5) | 0.68 | 0.24–1.91 | 0.469 | 0.36 | 0.07–1.85 | 0.225 |
| Farmer | 2 (7.7) | 24 (92.3) | 0.61 | 0.13–2.80 | 0.529 | 0.55 | 0.05–5.82 | 0.625 |
| Labor | 5 (7.0) | 66 (93.0) | 0.55 | 0.20–1.56 | 0.266 | 0.38 | 0.06–2.25 | 0.290 |
| **Having regular income** | | | | | | | | |
| No | 24 (10.1) | 213 (89.9) | 1.00 | | | 1.00 | | |
| Yes | 7 (9.0) | 71 (91.0) | 1.14 | 0.47–2.76 | 0.767 | 1.41 | 0.29–6.79 | 0.669 |
| **Smoking** | | | | | | | | |
| No | 15 (6.6) | 212 (93.4) | 1.00 | | | 1.00 | | |
| Yes | 16 (18.2) | 72 (81.8) | 3.14 | 1.47–6.67 | <0.003* | 1.94 | 0.55–6.79 | 0.295 |
| **Alcohol use** | | | | | | | | |
| No | 13 (7.4) | 162 (92.6) | 1.00 | | | 1.00 | | |
| Yes | 18 (12.9) | 122 (87.1) | 1.83 | 0.86–3.89 | 0.112 | 0.48 | 0.15–1.55 | 0.222 |
| **Methamphetamine use** | | | | | | | | |
| No | 26 (8.9) | 267 (91.1) | 1.00 | | | 1.00 | | |
| Yes | 5 (22.7) | 17 (77.3) | 3.02 | 1.03–8.85 | 0.044* | 1.83 | 0.30–11.05 | 0.508 |
| **Heroin use** | | | | | | | | |
| No | 28 (9.2) | 278 (90.8) | 1.00 | | | 1.00 | | |
| Yes | 3 (33.3) | 6 (66.7) | 4.96 | 1.17–20.93 | 0.029* | 15.40 | 0.11–2044.62 | 0.273 |
| **Crystal methamphetamine use** | | | | | | | | |
| No | 29 (9.4) | 278 (90.6) | 1.00 | | | 1.00 | | |
| Yes | 2 (25.0) | 6 (75.0) | 3.19 | 0.61–16.56 | 0.166 | 0.12 | 0.01–22.58 | 0.431 |

*(Continued)*

**Table 4.** (Continued)

| Factors | Early sexual initiation | | OR | 95% CI | p value | AOR | 95% CI | p value |
|---|---|---|---|---|---|---|---|---|
| | Yes n (%) | No n (%) | | | | | | |
| **Work experience outside village** | | | | | | | | |
| No | 20 (9.0) | 203 (91.0) | 1.00 | | | 1.00 | | |
| Yes | 11 (12.0) | 81 (88.0) | 1.37 | 0.63–3.00 | 0.420 | 2.21 | 0.73–6.66 | 0.156 |
| **Work experience abroad** | | | | | | | | |
| No | 29 (13.4) | 273 (90.4) | 1.00 | | | 1.00 | | |
| Yes | 2 (15.4) | 11 (84.6) | 1.71 | 0.36–8.10 | 0.498 | 4.03 | 0.42–38.18 | 0.224 |
| **Level of knowledge of safe sex** | | | | | | | | |
| Poor | 25 (11.6) | 191 (88.4) | 2.48 | 0.31–19.38 | **0.385** | 1.90 | 0.11–32.08 | **0.656** |
| Moderate | 5 (6.3) | 74 (93.7) | 1.28 | **0.14–11.64** | 0.824 | 1.28 | **0.06–25.32** | 0.869 |
| Good | 1 (5.0) | 19 (95.0) | **1.00** | | | **1.00** | | |
| **Level of attitudes toward early sexual initiation behaviors and safe sex** | | | | | | | | |
| Poor | 13 (12.0) | 95 (88.0) | 1.94 | 0.66–5.70 | 0.226 | 0.76 | 0.18–3.17 | 0.715 |
| Moderate | 13 (9.9) | 118 (90.1) | 1.56 | 0.53–4.57 | 0.414 | 1.10 | 0.28–4.32 | 0.883 |
| Good | 5 (6.6) | 71 (93.4) | 1.00 | | | 1.00 | | |

*Significance level at α =0.05.

dominance of males [35] and the culture of early marriage among hill tribes could lead to early sexual initiation.

Our study revealed that younger hill tribe youths are more likely to have early sexual initiation than older youths are. Our study includes an almost 10-year interval of participants between 15 and 24 years of age, followed by a wide range of ages. Younger individuals might be exposed to new technology and mass media of pornographic materials and seek partners, which leads them to have early sexual initiation. Lay et al. [40] reported that younger youths, both boys and girls, were associated with early sexual initiation among Brazilian adolescents.

Several substance use behaviors, including smoking, methamphetamine use, and heroin use, are associated with early sexual initiation among hill tribe youths. Exposure to substances could lead to uncontrolled individuals' emotions, particularly those of young age, and, ultimately, sexual intercourse. A systematic review of cohort studies reported that youths' substance use behaviors were strong predictors of early sexual initiation [41]. A study conducted in the United States reported that substance use increased early sexual initiation [42]. Substance use was a significant predictor of early sexual initiation among young people in Brazil [43] and Kenya [44].

In our study, 83 participants (6.3%) did not respond to the question asking about sexual initiation experience. Tamorpark et al. [45] reported that asking a sensitive question, particularly about sexual behavior and substance use, in an audio computer-assisted self-interview was more accurate than a paper-based questionnaire. Given that hill tribe people are not familiar with discussing sexual behavior with anyone, this is not an overestimate for researchers. In several traditional cultures, especially marriage in early life [35] and parenting styles in which sexual expression is rarely discussed among family members [20], young people tend to learn sexual education from the media. Moreover, several hill tribe youths move to the city for education or work, with the opportunity to be exposed to sexual initiation at an early age. The major driving factors that contribute to hill tribe youths' exposure to early sexual initiation are poverty and limited literacy in both their parents and themselves.

A few limitations were found during the study. Many questions on the study protocol inquire about sexual behaviors; however, some tribes do not commonly talk about sexual behavior in their culture, particularly in public areas. In our study, 18 Akha women, 14 Lahu women, and 12 Yao women did not respond to these questions. Therefore, the findings on sexual behaviors could be underestimated in some aspects; however, the researchers explained to the participants as thoroughly as possible that their information would remain private and confidential and that it could not be used to identify the participants later on. During the process of data collection, all essential information, including concerns regarding privacy and confidentiality, was carefully explained to all participants to increase their willingness to answer the questions truthfully. The participants cooperated well with this aspect of the study. Moreover, the number of participants in each tribe was different, which could lead to the generalizability of the findings to a specific tribe. Finally, a few participants did not clearly understand the context of some of the questions due to limited literacy. To address this issue, the researchers personally explained the meaning and content of confusing items. This study has various strengths, such as the concept of investigating the very sensitive issue of early sexual initiation among the marginalized population among six main hill tribes and the large number of participants recruited to the study.

## Conclusion

Hill tribe youths, particularly male youths, face the problem of early sexual initiation. Several substances contribute to early sexual initiation among hill tribe youths. Therefore, interventions to reduce early sexual initiation behavior among hill tribe youths should focus on male gender and substance use. Understanding the context, social norms and culture of sexual activity and the ability to discuss the issue at the family level are also critical points. Sexual education interventions should be implemented among family members, community-oriented programs, and extracurricular activities to supplement formal education in Thailand.

## Supporting information

**S1 File. Questionnaire.** A validated questionnaire consisted of four parts with a total of 73 questions.
(PDF)

**S2 File. Study data.** Excel data from the study with codes that were used for the analysis.
(XLSX)

## Acknowledgments

We would like to thank all the village headmen for their kind support in accessing the participants. We would also like to thank all the participants for providing essential information in the study.

## Author contributions

**Conceptualization:** Ratipark Tamornpark, Tawatchai Apidechkul.

**Data curation:** Ratipark Tamornpark, Tawatchai Apidechkul, Panupong Upala, Chalitar Chomchoei, Fartima Yeemard.

**Formal analysis:** Ratipark Tamornpark, Tawatchai Apidechkul.

**Funding acquisition:** Tawatchai Apidechkul.

**Investigation:** Ratipark Tamornpark, Tawatchai Apidechkul, Panupong Upala, Chalitar Chomchoei, Fartima Yeemard.

**Methodology:** Ratipark Tamornpark, Tawatchai Apidechkul.

**Project administration:** Tawatchai Apidechkul.

**Supervision:** Ratipark Tamornpark, Tawatchai Apidechkul.

**Writing – original draft:** Ratipark Tamornpark, Tawatchai Apidechkul, Panupong Upala, Chalitar Chomchoei, Fartima Yeemard.

**Writing – review & editing:** Ratipark Tamornpark, Tawatchai Apidechkul, Panupong Upala, Chalitar Chomchoei, Fartima Yeemard.

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
