## [Decision Letter · Decision Letter 0]

20 Mar 2024

PONE-D-23-33898Influencing factors early sexual initiation among hill tribe youths in northern ThailandPLOS ONE

%LAST_Dear Dr. Apidechkul,

Thank you for submitting your manuscript to PLOS ONE. After careful consideration, we feel that it has merit but does not fully meet PLOS ONE’s publication criteria as it currently stands. Therefore, we invite you to submit a revised version of the manuscript that addresses the points raised during the review process.

We look forward to receiving your revised manuscript.

Kind regards,

George Kuryan

Academic Editor

PLOS ONE

Journal Requirements:

3. Thank you for stating the following financial disclosure: "The study was funded by The National Research Council of Thailand and Mae Fah Laung University, Thailand (Grant No.06/2560). " 

4. In the online submission form, you indicated that the data underlying the results presented in the study are available by contacting directly to the corresponing author.

Reviewers' comments:

Reviewer's Responses to Questions

**Comments to the Author**

1. Is the manuscript technically sound, and do the data support the conclusions?

Reviewer #1: Partly

2. Has the statistical analysis been performed appropriately and rigorously? 

Reviewer #1: No

3. Have the authors made all data underlying the findings in their manuscript fully available?

Reviewer #1: Yes

4. Is the manuscript presented in an intelligible fashion and written in standard English?

Reviewer #1: No

5. Review Comments to the Author

Reviewer #1: See attached file.

6. PLOS authors have the option to publish the peer review history of their article (what does this mean? ). If published, this will include your full peer review and any attached files.

**Do you want your identity to be public for this peer review?** For information about this choice, including consent withdrawal, please see our Privacy Policy .

Reviewer #1: No

---

## [Author Response · Author response to Decision Letter 1]

20 Apr 2024

Repose to reviewers and editor comments

Dear Editor of Ploseone,

Thank you very much or the opportunity to improve our paper with excellent comments from reviewer. We have carefully revised by re-analysis in the whole process based in the conditions suggested by reviewers.

TK

PONE-D-23-33898

Influencing factors early sexual initiation among hill tribe youths in northern Thailand

PLOS ONE

Dear Editor,

Thank you very much for opportunity to review the manuscript titled " Influencing factors early sexual initiation among hill tribe youths in northern Thailand" for PLOS ONE.

I think that some aspects of this manuscript are interesting. I believe that some major changes are required. Below are my comments about this manuscript, which I hope are some assistances to the authors.

1.The title, author should be specific “in Chiang Rai province”? not representative in northern of Thailand, and may considering add approach “: community based cross sectional approach”

: Thank you, it’s improved

2.An abstract

- Author conclusions “A large proportion of the hill tribe youths have experienced early sexual initiation before the age of 15 years. A specific training program should be integrated into the high school study curriculum to protect early sexual initiation behaviors and safe sex, particularly in males. The curriculum should focus on the impact of alcohol and working in unskilled work sectors and potentially early sexual initiation. As the result based author may conclude in community or family based intervention, attitude come from socialization in family….etc. Thailand has a sexual education curriculum at all level in school.

: Thank you for the comment. We know that Thailand has a curriculum at all school levels, but we intend to add an extra and special curriculum. You said that it exists at all levels, but according to the findings of our program, those people still lack skills and knowledge in safe sex. This is the reason that we put this statement together. We apologize if this makes you confused.

: We also agree with you to add family and community levels, which we have added information in the section.

2. Methodology,

- page 5 line 25-27 author mention that “In the section on knowledge of safe sex (10 items), there were classified into three categories; poor (0-5 scored), moderate (6-8 scored), and good (9-10 scored).” It is knowledge about HIV transmission (not safe sex). Please clearly how scoring and range of score and cutting point? Including reference?

: This category is formed by the findings from our previous study in Ref. No.23 and also on our pilot phase. It’s improved.

-Substance use question between Thai and English inconsistent. The answer means? Current use? Ever but quit? Never or ever tried? Please clearly explain at variable section. What yes?

: Thank you for the comment. Based on our developed initial questionnaire, three answers to these questions were provided. However, after piloting, two options were found: better validity and reliability. Then, we used two options to answer these questions. We have revised the questionnaire. Thank you so much for the great comment, which is excellent for those who will apply our original questionnaire in the future. Our update/revised questionnaires (ENG and THAI) are attached.

3. Results

- Table 3 Something error? Author present variable” Alcohol or drug use prior having sexual initiation with 3 answers <no, sometime, yes> but in questionnaire part 3 item 3.9 in English and Thai language the answer only <Yes, No>

: Please see our response in your previous comment.

- Table 1 and 2 author should present as below

characteristics Total Early sexual initiation Never had sexual experience � 2 p-value

Yes (<14) No (�15)

n % n % n % n %

: Thank you so much for your great comment, we agreed with you and have carefully revised tables, please see our new tables 1, 2, and 4

-Table 4 Author definition “Early sexual initiation” yes= age at first sexual initiation <14 and No= age at first sexual initiation �15 including never had sexual experience. Author should exclude never had sexual experience or use as reference.

: We thank you so much for such great suggestion!. We have re-run the model by comparing those who had sexual before and after 15 years by excluding those who did not have sexual experience from the model. Please see detail in table 4. The new model really reflects the objective of the study.

4. Discussion

Page 15-line 16-18 Author discuss that “The hill tribe youths have a younger age of their first sexual initiation than Thai youths which might be the culture of the hill tribe that needs to get married early to have a child for help in their family work.” Is it culture issue? Economic or occupational Issue? Author should explain how culture difference from hill tribe and general youths.

: Thank you for your feedback. It’s added the culture issue on page 19-line 9-20.

Thank you so much,

TK

Associate Prof. Dr. Tawatchai Apidechkul, MSc (Infectious Epidemiology), Dr.P.H. (Epidemiology)

School of Health Science, Mae Fah Luang Univeristy, Thailand

Director, Center of Excellence of Hill Tribe Health Research

Former Hubert H Humphrey Fellow (2013-2014), Emory University

Candidate MMSc-GHD, Harvard Medical School, Harvard University, United States

---

## [Decision Letter · Decision Letter 1]

13 Aug 2024

PONE-D-23-33898R1Influencing factors early sexual initiation among hill tribe youths in Chiang Rai Province, northern Thailand: A community-based cross-sectional studyPLOS ONE

Dear Dr. Apidechkul,

Thank you for submitting your manuscript to PLOS ONE. After careful consideration, we feel that it has merit but does not fully meet PLOS ONE’s publication criteria as it currently stands. Therefore, we invite you to submit a revised version of the manuscript that addresses the points raised during the review process.

We look forward to receiving your revised manuscript.

Kind regards,

George Kuryan

Academic Editor

PLOS ONE

Journal Requirements:

Reviewers' comments:

Reviewer's Responses to Questions

**Comments to the Author**

1. If the authors have adequately addressed your comments raised in a previous round of review and you feel that this manuscript is now acceptable for publication, you may indicate that here to bypass the “Comments to the Author” section, enter your conflict of interest statement in the “Confidential to Editor” section, and submit your "Accept" recommendation.

Reviewer #2: All comments have been addressed

2. Is the manuscript technically sound, and do the data support the conclusions?

Reviewer #2: Yes

3. Has the statistical analysis been performed appropriately and rigorously? 

Reviewer #2: Yes

4. Have the authors made all data underlying the findings in their manuscript fully available?

Reviewer #2: Yes

5. Is the manuscript presented in an intelligible fashion and written in standard English?

Reviewer #2: Yes

6. Review Comments to the Author

Reviewer #2: Reviewers’ comments

Title: Influencing factors early sexual initiation among hill tribe youths in Chiang Rai Province, northern Thailand: A community-based cross-sectional study

The manuscript was well-prepared. I have a few comments as follows.

Abstract

-The conclusion did not come from the study findings. The authors can be more specific, like suggesting what to be include in the training program (what knowledge is lacking from the results).

Methods

-The authors mentioned that the questionnaires were piloted in Mae Chan district. Where did the study take place? Please add in the study setting.

-Please clarify about the recruitment which should happen without coercion. The authors stated that they contacted village headmen for their permission to access the villages, but how those youth was reached? As most of them were studying, were they in the village or lived somewhere outside the village for their schooling? Could any youth deny participation? Did their parents, family members, and other villagers know about their entering the study? How many of them refused to join? Any compensation to youth, their family, or villager headmen?

-Please describe how youth were instructed before completing the questionnaires. Were there any specific guide written in the questionnaire or mentioned verbally, like the time frame for alcohol, smoking, and substance use (lifetime, or any specific duration to be reported), definition of sexual intercourse, etc.)

Results

Table 1: The column “Early sexual initiation” can list only “yes” (and add the denominator somewhere). Similarly for variables with yes/no response, presentation in the table can be only yes which would make them easier to read and reduce the table size.

Discussion

Page 18 line 17-18. The study found that a larger proportion of male had early sex initiation than female youth. Was it a result of reporting bias? Males perceived power over females when they initiated sex, while females might have early sex without their willingness. Thus, males were more likely to report having early sex. It might be related to patriarchy in Thai society, which could be prominent in hill tribe cultures. Just a thought, I know the authors are experts on this issue. Could the authors provide any comments or discussion on this possibility.

Page 19 line 32” the term “poor education” might not appropriate and too broad, given that most participants had attended school. The authors could consider changing to other words like “limited literacy”, or “low reading ability”.

7. PLOS authors have the option to publish the peer review history of their article (what does this mean? ). If published, this will include your full peer review and any attached files.

**Do you want your identity to be public for this peer review?** For information about this choice, including consent withdrawal, please see our Privacy Policy .

Reviewer #2: **Yes: ** Linda Aurpibul

---

## [Author Response · Author response to Decision Letter 2]

22 Aug 2024

Response to the reviewer comment

PONE-D-23-33898R1

Influencing factors early sexual initiation among hill tribe youths in Chiang Rai Province, northern Thailand: A community-based cross-sectional study

PLOS ONE

Dear Editor and Reviewers,

Thank you very much for the opportunity to revise that paper. We have carefully and completely revised your comments. The paper also double-checked the grammatical error before re-submitting. Please kindly the following our responses.

Thank you,

TK

+++

Reviewer #2: Reviewers’ comments

Title: Influencing factors early sexual initiation among hill tribe youths in Chiang Rai Province, northern Thailand: A community-based cross-sectional study

The manuscript was well-prepared. I have a few comments as follows.

Abstract

-The conclusion did not come from the study findings. The authors can be more specific, like suggesting what to be include in the training program (what knowledge is lacking from the results).

Methods

: It is a sexual education focused on safe sex and early sexual interaction protection.

-The authors mentioned that the questionnaires were piloted in Mae Chan district. Where did the study take place? Please add in the study setting.

: Thank you. It’s added which are Huan Hin Fon and Sun Ti Suk Villages

-Please clarify about the recruitment which should happen without coercion. The authors stated that they contacted village headmen for their permission to access the villages, but how those youth was reached? As most of them were studying, were they in the village or lived somewhere outside the village for their schooling? Could any youth deny participation? Did their parents, family members, and other villagers know about their entering the study? How many of them refused to join? Any compensation to youth, their family, or villager headmen?

: There was nobody to refuse the participants. In the process of recruitment, after providing them with information about the study, all participants were informed that they had the right to refuse or deny joining the project as the standard protocol for human research ethical procedure. All parents were also informed about the study, and only those who had children aged less than 17 and below were asked for their permission to gather the information from the participants (youths). All participants and parents agreed and were willing to join the project. During the study, we did not have any inclusion and exclusion criteria for whether the participants were in or out of the school. All participants were compensated 200 baht ($6), which is clearly declared in the ethical consideration process.

-Please describe how youth were instructed before completing the questionnaires. Were there any specific guide written in the questionnaire or mentioned verbally, like the time frame for alcohol, smoking, and substance use (lifetime, or any specific duration to be reported), definition of sexual intercourse, etc.)

: Each question provided complete and clear information. We did not have any specific guide for completing the questionnaire; however, all participants were told that if there was any confusion, they could contact the researcher. Nobody asked to clear any questions.

: Since we worked a cross-sectional, then the questions were asked directly with the time frame, except for some question that we had a clear its frame in the questions.

Results

Table 1: The column “Early sexual initiation” can list only “yes” (and add the denominator somewhere). Similarly for variables with yes/no response, presentation in the table can be only yes which would make them easier to read and reduce the table size.

: Thank you for the comments; however, after discussing with the team, we decided to maintain this column and information to make it easier and quicker for the readers to go through the details.

Discussion

Page 18 line 17-18. The study found that a larger proportion of male had early sex initiation than female youth. Was it a result of reporting bias? Males perceived power over females when they initiated sex, while females might have early sex without their willingness. Thus, males were more likely to report having early sex. It might be related to patriarchy in Thai society, which could be prominent in hill tribe cultures. Just a thought, I know the authors are experts on this issue. Could the authors provide any comments or discussion on this possibility.

: Thank you for this great idea. We had discussed this point while making the conclusion; however, with the study design, we could not extend our thoughts outside of what we found in the results. One of our team is trained in medical anthropology and is trying to work on her project to understand some critical issues from our findings, particularly gender roles and sexual inequality in some tribes. We very much hope that we will be able to publish our findings soon.

Page 19 line 32” the term “poor education” might not appropriate and too broad, given that most participants had attended school. The authors could consider changing to other words like “limited literacy”, or “low reading ability”.

: Thank you very much for the suggestion; we agree. It’s changed it to limited literacy.

Thank you very much,

TK

Assoc Prof. Dr. Tawatchai Apidechkul, MSc (Infectious Epidemiology), Dr. P. H (Epidemiology)

Director, Center of Excellence of Hill Tribe Health Research,

School of Health Science, Mae Fah Luang University, Thailand

Former Hubert H Humphrey Fellow (2013-2014), Emory University

---

## [Decision Letter · Decision Letter 2]

29 Jan 2025

PONE-D-23-33898R2Influencing factors early sexual initiation among hill tribe youths in Chiang Rai Province, northern Thailand: A community-based cross-sectional studyPLOS ONE

Dear Dr. Apidechkul,

Thank you for submitting your manuscript to PLOS ONE. After careful consideration, we feel that it has merit but does not fully meet PLOS ONE’s publication criteria as it currently stands. Therefore, we invite you to submit a revised version of the manuscript that addresses the points raised during the review process.

We look forward to receiving your revised manuscript.

Kind regards,

Ghulam Yaseen, Ph.D.

Academic Editor

PLOS ONE

Journal Requirements:

Reviewers' comments:

Reviewer's Responses to Questions

**Comments to the Author**

1. If the authors have adequately addressed your comments raised in a previous round of review and you feel that this manuscript is now acceptable for publication, you may indicate that here to bypass the “Comments to the Author” section, enter your conflict of interest statement in the “Confidential to Editor” section, and submit your "Accept" recommendation.

Reviewer #2: All comments have been addressed

Reviewer #3: All comments have been addressed

2. Is the manuscript technically sound, and do the data support the conclusions?

Reviewer #2: Yes

Reviewer #3: Yes

3. Has the statistical analysis been performed appropriately and rigorously? 

Reviewer #2: Yes

Reviewer #3: Yes

4. Have the authors made all data underlying the findings in their manuscript fully available?

Reviewer #2: Yes

Reviewer #3: Yes

5. Is the manuscript presented in an intelligible fashion and written in standard English?

Reviewer #2: No

Reviewer #3: No

6. Review Comments to the Author

Reviewer #2: Reviewers’ comments

Thank you for the revised manuscript. The authors have done a great job in improving it.

I have only a few comments as follows.

Abstract

Page 2 line #3-4: males was compared to males (one should be females?)

Page 2 line# 12: impact of substance use was mentioned in the conclusion, but no information in the results of abstract to explain how they were related. I understand it later when reading results part in the main text. Could the authors add a sentence about substance use in the results to guide the readers that the conclusion did not come from nowhere?

Methods

Page 4 line#15: “…, it needed 1293 samples in the study” What is it refer to? Should the author modify it to something like “we needed 1293 participants in the study”

Page 4 line#29: “Those questions that scored less than 0.5 were excluded.” I think it should be “less than or equal to 0.50”, isn’t it? Please recheck.

Results

Page 7 line # 14: It seemed like the word “difference” is missing from the 3rd paragraph. What was statistically significant is the difference, isn’t it? Please check.

Page 13 line#3-4.: the percentage was different from other place (13.6% had sexual initiation before 15 years of age”, it was 9.8% elsewhere). Please recheck.

Page 13 line#13-17: P-value in the text should be written only “p”, usually small p with italic.

Discussion

Page 17 line#21: I think it should be “Males and younger age…” not females.

Page 18 line # 31-33: The sentence is confusing, please revise the language.

Page 19 line# 9: the phrase “, it was found that” could be removed as it has no meaning in the sentence.

Page 19 line#17-18: Please add numbers from the study finding to support the statement as the authors did in other statements.

Language quality

Overall, the language in this article is difficult to read, not flow while reading. I would advise the article to be edited by native English speaker for quality improvement.

Reviewer #3: I can see the authors have improved previous versions of the manuscript, based on reviewers' comments. The study is of importance because there are important results that should be known in the country and other countries with similar situations. However, despite improvements, there are several places where the descriptions and language used need to be revised in order to make it clear for the reader. I have made a number of suggestions to edit the document in the attached file. Please after carrying out corrections have the document checked by someone who speaks and writes English, so it is fully edited. Thanks.

7. PLOS authors have the option to publish the peer review history of their article (what does this mean? ). If published, this will include your full peer review and any attached files.

**Do you want your identity to be public for this peer review?** For information about this choice, including consent withdrawal, please see our Privacy Policy .

Reviewer #2: **Yes: ** Linda Aurpibul

Reviewer #3: **Yes: ** Dr. Alfredo L. Fort

---

## [Author Response · Author response to Decision Letter 3]

27 Feb 2025

Response to reviewer comments

Reviewer #2: Reviewers’ comments

Thank you for the revised manuscript. The authors have done a great job in improving it.

I have only a few comments as follows.

Abstract

Page 2 line #3-4: males was compared to males (one should be females?)

: We apologize for this mistake; it should read female. Please see line 6, page 2.

Page 2 line# 12: impact of substance use was mentioned in the conclusion, but no information in the results of abstract to explain how they were related. I understand it later when reading results part in the main text. Could the authors add a sentence about substance use in the results to guide the readers that the conclusion did not come from nowhere?

Methods

: Thank you for this helpful comment. This occurs because the statistics are significant in the univariate analysis and chi-square test. However, in the multivariable analysis, there were no associations with early sex. Thus, we slightly extended our thoughts in the Conclusion sections. We have also added a sentence to the Abstract (page 2, lines 2-3).

Page 4 line#15: “…, it needed 1293 samples in the study” What is it refer to? Should the author modify it to something like “we needed 1293 participants in the study”

: Thank you; it has been revised. Please see page 4, lines 12-13.

Page 4 line#29: “Those questions that scored less than 0.5 were excluded.” I think it should be “less than or equal to 0.50”, isn’t it? Please recheck.

: Sorry for the mistake, it is 0.5. We have revised the sentence in the text as follows: “Those questions that scored less than 0.5 were excluded, questions that scored 0.50-0.69 were revised as comments…”

Results

Page 7 line # 14: It seemed like the word “difference” is missing from the 3rd paragraph. What was statistically significant is the difference, isn’t it? Please check.

: Thank you; this information was added. Please see page 7, line 14.

Page 13 line#3-4.: the percentage was different from other place (13.6% had sexual initiation before 15 years of age”, it was 9.8% elsewhere). Please recheck.

: Thank you so much. This small error prompted us to check the entire paper. Now, all the points are correct.

Page 13 line#13-17: P-value in the text should be written only “p”, usually small p with italic.

Discussion

Thank you for the comment. We have discussed this point, and we have to maintain its original form because two of our statisticians inform us that there is a specific requirement for the journal. However, using the full term is also common, as used in our previous papers. We hope that you understand.

Page 17 line#21: I think it should be “Males and younger age…” not females.

: Thank you so much for calling this to our attention; we have corrected it accordingly.

Page 18 line # 31-33: The sentence is confusing, please revise the language.

: Thank you for the comment. These sentences were drafted by one of our coauthors during the manuscript development. However, we forgot to include the details of the reference. Once we read the whole paragraph, it still clearly presents the information we want to present. However, we deleted the sentences from the text. Thank you so much for pointing this out.

Page 19 line# 9: the phrase “, it was found that” could be removed as it has no meaning in the sentence.

: Thank you; this has been deleted.

Page 19 line#17-18: Please add numbers from the study finding to support the statement as the authors did in other statements.

: Thank you for this helpful comment. We have added this number to the paragraph. Please see page 19, lines 16-17.

Language quality

Overall, the language in this article is difficult to read, not flow while reading. I would advise the article to be edited by native English speaker for quality improvement.

: Thank you; the paper has been double-checked by AJE (certificate no: 2012-7831-AE6C-33FF-026E)

Reviewer #3: I can see the authors have improved previous versions of the manuscript, based on reviewers' comments. The study is of importance because there are important results that should be known in the country and other countries with similar situations. However, despite improvements, there are several places where the descriptions and language used need to be revised in order to make it clear for the reader. I have made a number of suggestions to edit the document in the attached file. Please after carrying out corrections have the document checked by someone who speaks and writes English, so it is fully edited. Thanks.

: Thank you. The paper has been double-checked by AJE (certificate no: 2012-7831-AE6C-33FF-026E.).

1. Check the number/information in percentage of having had sexual experience.

The text has been checked and corrected; see page 1, lines 33-35.

2. Page 2, lines 27-28, delete “old; these behaviors vary in different societies frequently occur among youths [13]”

: This has been deleted per your suggestion.

3. Page 3, line 1, “lived”… “live”

: This has been deleted per your suggestion.

4. Page 3, lines 10, delete “the hill tribe youths who have been clearly defined as”

: This has been deleted per your suggestion.

5. Page 3, delete “then” in line18

6. Many points in the whole paper, in the PDF file, are asked to improved.

: All the points have been improved by AJE.

TK

Assoc Prof. Dr. Tawatchai Apidechkul, MSc (Infectious Epidemiology), Dr. P. H (Epidemiology)

Director, Center of Excellence of Hill Tribe Health Research,

School of Health Science, Mae Fah Luang University, Thailand

Former Hubert H Humphrey Fellow (2013-2014), Emory University

---

## [Editor Report · Decision Letter 3]

2 Mar 2025

Factors influencing early sexual initiation among hill tribe youths in Chiang Rai Province, northern Thailand: A community-based cross-sectional study

PONE-D-23-33898R3

Dear Dr. Apidechkul,

We’re pleased to inform you that your manuscript has been judged scientifically suitable for publication and will be formally accepted for publication once it meets all outstanding technical requirements.

Kind regards,

Ghulam Yaseen, Ph.D.

Academic Editor

PLOS ONE
---

## [Editor Report · Acceptance letter]

PONE-D-23-33898R3

PLOS ONE

Dear Dr. Apidechkul,

I'm pleased to inform you that your manuscript has been deemed suitable for publication in PLOS ONE. Congratulations! Your manuscript is now being handed over to our production team.

Kind regards,

on behalf of

Professor Ghulam Yaseen

Academic Editor

PLOS ONE